# Advanced Receiver Operating Characteristic Curve Analysis to Identify Outliers in Binary Machine Learning Classifications for Precision Medicine

Khashayar Namdar[1,2,4,7], Farzad Khalvati[1,2,3,4,5,6,7] *

[1]Department of Diagnostic & Interventional Radiology, The Hospital for Sick Children, Toronto, ON, Canada
[2]Neurosciences & Mental Health Research Program, SickKids Research Institute, Toronto, ON, Canada
[3]Department of Medical Imaging, University of Toronto, Toronto, ON, Canada
[4]Institute of Medical Science, University of Toronto, Toronto, ON, Canada
[5]Department of Computer Science, University of Toronto, Toronto, ON, Canada
[6]Department of Mechanical and Industrial Engineering, University of Toronto, Toronto, ON, Canada
[7]Vector Institute, Toronto, ON, Canada
* farzad.khalvati@utoronto.ca

*Abstract*—The Receiver Operating Characteristic (ROC) curve is a critical tool for binary classification analysis in medicine, with the Area Under the ROC Curve (AUROC) serving as a widely accepted metric to evaluate the performance of binary classifiers. This study conducts a comprehensive review of the ROC curve with a focus on its utility in outlier identification. We introduce a novel scoring method to rank actual positives and actual negatives within a test set, according to their impact on AUROC degradation. We bridge the scoring system with the ROC curve analysis to quantify each data point's contribution to AUROC loss. Furthermore, we introduce the IMICS ROC Analyzer, a graphical user interface-based software, embedded with our innovative algorithms. Through the use of an open-source prostate cancer dataset, we illustrate the application of our algorithms for practical outlier detection in binary classification tasks. The IMICS ROC Analyzer enhances the field of precision medicine by allowing for measuring an individual's contributions (be it patients, lesions, or samples) to the overall AUROC, thus facilitating confidence measurement of Machine Learning (ML) classifiers for individual cases of interest in a cohort.

*Index Terms*—AUROC, Binary Classification, Evaluation Metric, Machine Learning, Outlier Identification, ROC

## I. INTRODUCTION

Receiver Operating Characteristic (ROC) curve is a critical tool in binary classification evaluation, offering a visual demonstration of the balance between sensitivity and specificity across varying thresholds. ROC application spans multiple fields, notably in medical and engineering sciences, where it serves as an essential measure for assessing classifier efficacy.

The Area Under the ROC Curve (AUROC) represents a key metric associated with the ROC curve, holding particular significance within the medical field, although its adoption in engineering commenced in the early 1990s [1], [2]. Subsequent research on the AUROC has introduced several enhancements aimed at broadening its utility across diverse settings.

ROC and AUROC possess distinctive features that render them suitable for ML applications within the healthcare domain [3]. The ROC curve is instrumental in determining the optimal threshold for binary classifiers, such as linear models or Neural Networks (NN), enabling the conversion of predicted probabilities into class labels. Additionally, AUROC inherently balances the significance of each class, making it a valuable metric for evaluating the efficacy of binary classifiers in scenarios involving imbalanced datasets [4]. To the best of our knowledge, the majority of existing AUROC research has been concentrated on cohort-level analysis, with a notable absence of methodologies for evaluating the impact of individual contributions on AUROC loss and their influence on the ROC curve's configuration. Furthermore, there appears to be a lack of graphical user interface (GUI)-enabled software tailored for comprehensive ROC analysis. Our work aims to fill these gaps.

Key contributions of this research include:

- We implement an optimized Python function designed for plotting accurate and visually compelling ROC curves, and make it open source.
- We establish scoring metrics for ranking actual positives and actual negatives, helping to detect outliers within cohorts.
- We introduce a precise analytical methodology for quantifying the individual contributions of examples to AUROC loss within a specific ROC curve.
- We propose B point as an alternative to Youden's J point [5] for optimal ROC curve threshold selection.
- We develop and open-source the Intelligent Medical Informatics Computing Systems (IMICS) ROC Analyzer, a GUI-enabled software tool, to facilitate detailed ROC analysis.

## II. RELATED WORK

Kottas et al. contributed to the development of a methodology for estimating confidence intervals for the AUROC, thus providing a more dependable evaluation of classifier performance [6]. Expanding on this foundation, Yu et al. presented a revised AUROC metric tailored for gene ranking to support genetic research [7]. Yu and Park suggested a reformulated AUROC metric designed to penalize regression models in gene selection processes involving high-dimensional datasets [8].

The exploration of AUROC as a loss function for training ML models by Rosenfeld et al. showed that models trained on an AUROC basis exhibited superior generalization capabilities [9]. Their investigation, however, primarily focused on Support Vector Machines (SVM) [10] and did not include NN, particularly neglecting the incorporation of model confidence in the calculation of AUROC.

In online learning [11] contexts, Zhao et al. proposed an algorithm to maximize AUROC, thus facilitating effective adjustments to evolving data streams [12]. Ying et al. adopted a stochastic strategy, offering a different angle on AUROC optimization within online learning frameworks, which maximizes AUROC during training [13]. Cortes and Mohri studied the relationship between AUROC optimization and error rate reduction, elucidating that a decrease in error rate does not equal a decrease in AUROC [14].

While Ghanbari and Scheinberg focused on direct optimization of error rate and AUROC for classifiers [15], their methodologies were confined to linear classifiers. Namdar et al. proposed an augmented version of the AUROC that integrates the confidence of the model into its calculation [16]. Their findings indicated that this enhanced AUROC correlates with the Binary Cross Entropy (BCE) loss function, suggesting its superiority as a metric for supervising the training of Convolutional Neural Networks (CNNs). This methodology was validated across three distinct datasets, affirming its effectiveness. In this work, we use their fundamental AUROC calculation discussions.

Yan et al. discussed the inadequacies of traditional loss functions, including cross entropy (CE) and mean squared error (MSE), which are primarily aimed at improving classification accuracy but may not effectively optimize models for distinguishing between different outcomes at varying decision thresholds [17]. They introduced a novel objective function that serves as a differentiable surrogate for the Wilcoxon-Mann-Whitney (WMW) statistic [18]. This function is directly congruent with the AUROC, offering a more aligned approach to model optimization with respect to the AUROC metric. Inspired by this work, Namdar et al. developed an AUROC loss function in another research [19]. Instead of considering AUROC as the area under the ROC curve, they defined AUROC as a probability. This direct definition is a second pillar of our approach.

In contrast to the existing studies, our approach is focused on individual examples which are classified by a bi-

nary classifier. This aligns with the principles of precision medicine, facilitating the assessment of confidence levels in binary classifiers. Among the existing body of literature, the concept most akin to our approach is the identification of outliers. Evangelista et al. used rank distributions and fusing models for the classification of imbalanced data [20]. Nevertheless, their approach is based on pseudo-ROC curves that approximate the actual ROC curve. In contrast, our proposed method is empirical and inherently precise, lightweight, and faster. Additionally, our contributions are not limited to outlier identification, and we provide AUROC loss contribution for each example, all incorporated in a GUI-based software for comprehensive ROC analysis.

## III. METHODS

In the context of binary classification using an ML model, the output of the classifier is typically a probability score. This score, when applied against a threshold, is translated into a predicted class label, often represented as 0 or 1. Depending on the ground truth labels and the predicted labels, a true positive, true negative, false positive, or false negative can occur, which are shown by $TP$, $TN$, $FP$, and $FN$, respectively. The ROC represents the trade-off between sensitivity (also known as the true positive rate (TPR), described in Equation 1) and one minus specificity (Equation 2, the false positive rate (FPR)) across all potential thresholds ranging from 0 to 1.

$$TPR = \frac{TP}{TP + FN} \tag{1}$$

$$FPR = \frac{FP}{FP + TN} \tag{2}$$

Namdar et al. uncovered several characteristics of ROC curves [16]. They demonstrated that it is unnecessary to consider every conceivable value within this interval to effectively utilize thresholding. Instead, a limited set of *effective threshold boundaries* suffices, encompassing both the extreme values of 0 and 1 as well as the unique predicted probabilities. Additionally, they used actual positives (AP) and actual negatives (AN), to show examples with ground truth labels of 1 (APs), and examples with ground truth labels of 0 (ANs).

### A. IMICS ROC Plot Function

Inspired by previous works and Equations 1 and 2, we identified $AP$ number of rows with equal height on the vertical axis of an ROC curve and $AN$ number of columns with equal widths on its horizontal axis. This is fundamental for plotting an ROC curve. As a result, the grid for each ROC curve consists of rectangles with a width equal to $1/AN$ and a height of $1/AP$. Thus, an arbitrary setting of ticks on the X- and Y-axes is inappropriate.

AUROC equalizes class weights as if the number of APs and ANs were the same. To reflect this on the ROC plots, it is necessary to have a squared ROC curve, which is often neglected in the literature. Furthermore, the aesthetics of ROC curves in the literature are often inconsistent and inaccurate. To

bridge the gaps, we implemented and open-sourced a Python-based function called IMICS ROC plot (https://github.com/IMICSLab/ROC_Analyzer). IMICS ROC plot is compatible with Tufte's guidelines [21], which emphasize the use of minimal and clear visual designs with maximum data-ink and minimum non-data-ink ratios.

## B. AUROC Loss Scores

The AUROC can be efficiently calculated by comparing sets of APs and ANs that have been sorted by their predicted probabilities. Each example's contribution to AUROC loss is directly related to its position in this order, which can be shown using a rank. When examples are sorted by predicted probabilities, the first segment of examples whose number equals the size of ANs, contributes to AUROC accumulation if their ground truth label is 0; if not, they contribute to AUROC loss. For the subsequent segment, if their ground truth label is 0, it results in AUROC loss. Therefore, for the first examples in the sorted set, AUROC loss increases when lower-ranked examples (i.e., those with the lowest predicted probability) have a ground truth label of 1. Similarly, for the larger values in the sorted examples, more significant loss occurs when higher-ranked examples mistakenly have a ground truth label of 0.

The mathematical representation involves several key elements. Firstly, a set of sorted predicted probabilities, $SPr$, and a corresponding set of ground truth labels, $GT$ are provided. $n$ is defined as the total number of examples in the cohort, with each example $i$ having a predicted probability, $pr_i$, as shown in Equation 3. The true label for each example $i$ is denoted by $y_i$ in Equation 4. The dataset is composed of $n_p$ APs and $n_n$ ANs, as outlined in Equation 5.

$$SPr = \{pr_i \mid pr_i \in [0,1], \text{ for } i = 1, 2, \ldots, n\} \quad (3)$$

$$GT = \{y_i \mid y_i \in \{0,1\}, \text{for } pr_i \in SPr \text{ and } i = 1, 2, \ldots, n\} \quad (4)$$

$$n = n_n + n_p \quad (5)$$

The set $R$ is introduced to represent the rank of each example (Equation 6). The influence of an example on the overall AUROC metric, $E$, is specified in Equation 7. To assess the magnitude of the impact, the severity of this influence is mathematically defined in Equation 8, as $S$. This quantifies the potential effect an example can have on the total AUROC. Finally, the AUROC loss score is encapsulated in $SScore$, as detailed in Equation 9. A higher $SScore$ corresponds to a greater AUROC loss.

$$R(i) = i \quad \text{for } pr_i \in SPr \text{ and } i = 1, 2, \ldots, n \quad (6)$$

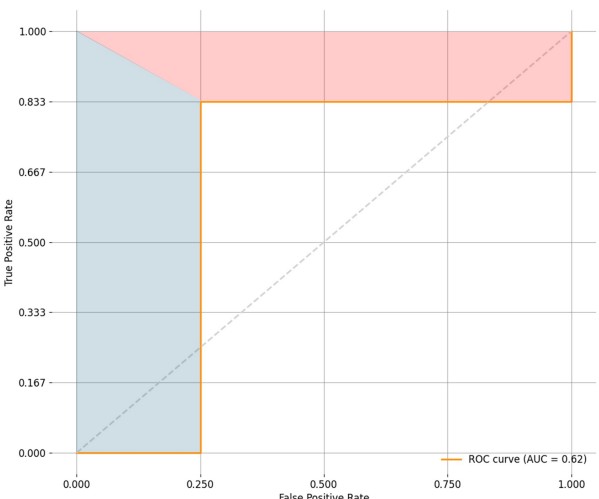

Fig. 1. Visualizing AUROC loss contributions

$$E(i) = \begin{cases} + & \text{if } i \leq n_n \text{ and } GT(i) = 0, \\ - & \text{if } i \leq n_n \text{ and } GT(i) = 1, \\ + & \text{if } i > n_n \text{ and } GT(i) = 1, \\ - & \text{otherwise.} \end{cases} \quad (7)$$

$$S(i) = \begin{cases} n_n - R(i) & \text{if } i \leq n_n, \\ n - R(i) & \text{if } i > n_n. \end{cases} \quad (8)$$

$$SScore(i) = \begin{cases} 0 & \text{if } E(i) = +, \\ S(i) \cdot \frac{1}{n_p} & \text{if } E(i) = - \text{ and } GT(i) = 1, \\ S(i) \cdot \frac{1}{n_n} & \text{if } E(i) = - \text{ and } GT(i) = 0. \end{cases} \quad (9)$$

## C. AUROC Loss Contributions for Individual Examples

The cumulative AUROC loss for a specified ROC curve is represented by the area above the curve. For ANs, the AUROC loss attributed to each individual example manifests itself as a vertical column, while for APs, it is represented as a horizontal row, as shown in Fig. 1. At the intersection points on the diagonal of the ROC grid, the AUROC contributions from APs and ANs overlap. These intersections are divided equally along their diagonals, facilitating the calculation of individual AUROC loss contributions for each example. Transitioning to the area beneath the ROC curve allows calculating the AUROC gain contribution for each individual example.

## D. B Point

We refer to the intersection of APs and ANs' AUROC loss contributions as the B line, also termed the label balance line. The point where the B line intersects with the ROC curve is referred to as the B point. B point signifies the juncture at which the weights of the two classes are balanced, as illustrated in Fig. 2.

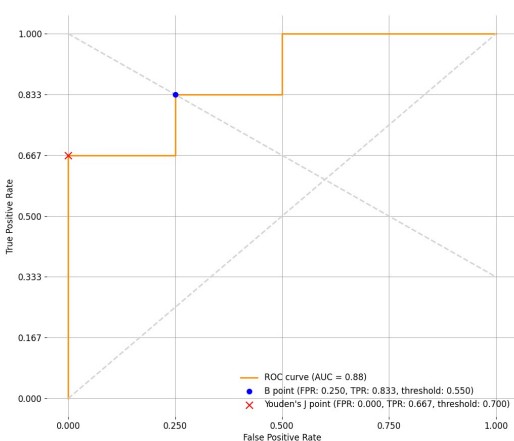

Fig. 2. B line and B point on ROC curves

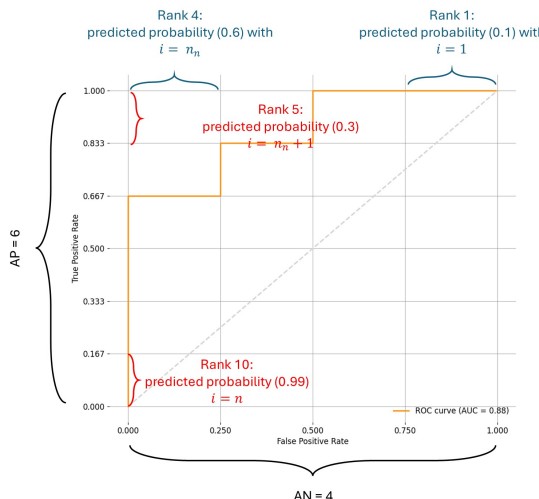

Fig. 4. Example 1 for ROC analysis

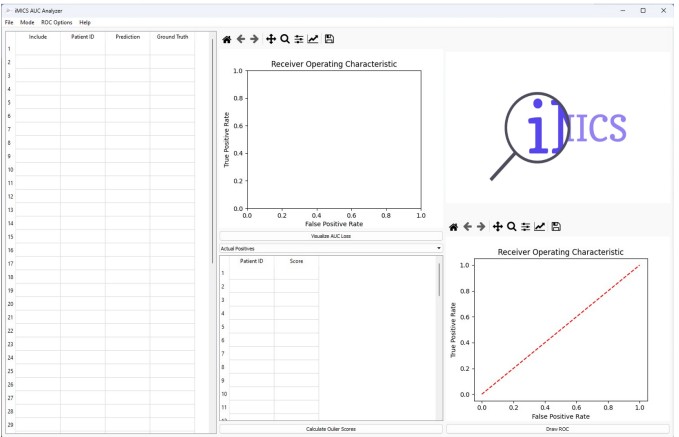

Fig. 3. IMICS ROC Analyzer

TABLE I
EXAMPLE 1 FOR ROC ANALYSIS

| True Label/Model Prediction | ANs | APs |
|---|---|---|
| GT | 0,0,0,0 | 1,1,1,1,1,1 |
| SPr | 0.1, 0.15, 0.5, 0.6 | 0.3, 0.55, 0.7, 0.8, 0.85, 0.99 |

### E. IMICS ROC Analyzer

To consolidate the proposed modules and enable their application in research, we developed and open-sourced a software named IMICS ROC Analyzer (https://github.com/IMICSLab/ROC_Analyzer). This software, built on PyQT [22] and featuring a GUI, includes individual example-based functionalities for calculating outlier scores and AUROC loss contributions (Fig. 3). Additionally, IMICS ROC Analyzer includes an investigation mode for excluding outliers and recalculating adjusted ROC and AUROC.

## IV. RESULTS

### A. Synthetic ROC

A synthetic ROC curve, shown in Fig. 4 and derived from Table I, illustrates the distribution of actual APs and ANs along the axes. It also demonstrates how AP and AN ranks influence contributions to the AUROC. The IMICS ROC Plot Function offers features for visualizing critical points such as Youden's J point and B point, along with the B line. These elements are detailed with their respective coordinates and threshold values on the ROC, as depicted in Fig. 5.

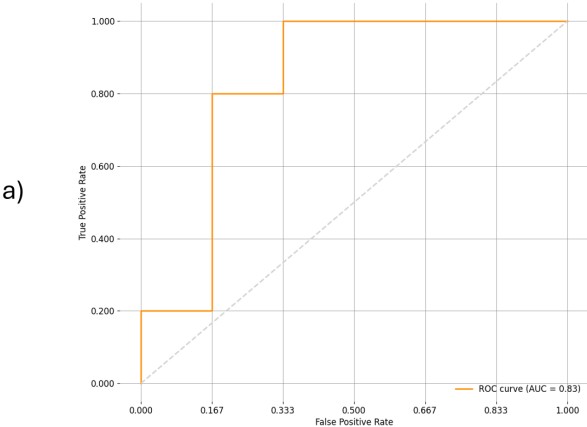

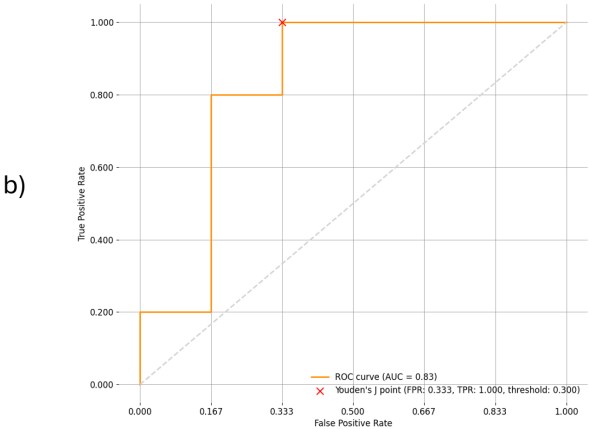

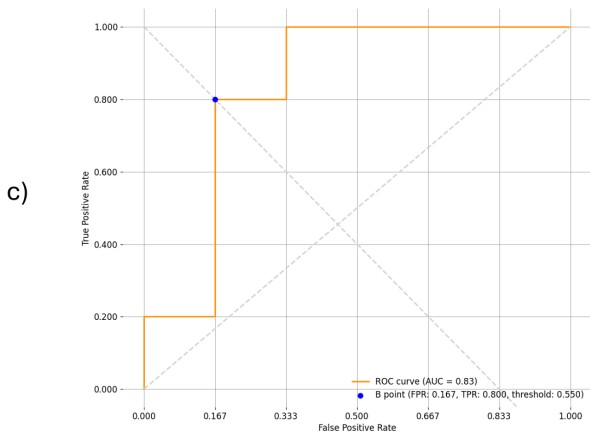

Fig. 5. Options of IMICS ROC Plot Function: a) basic ROC plot b) ROC with J point shown c) ROC with B line and B point shown

## B. ProstateX Dataset

To illustrate the application of our methodology to real-world datasets with more extensive examples, we chose the PROSTATEx dataset, sourced from the SPIE-AAPM-NCI ProstateX Challenge, which is a repository designed to advance the development and validation of image analysis methods for prostate cancer detection and characterization [23]. The training cohort of the dataset comprises multi-parametric Magnetic Resonance Imaging (mpMRI) scans obtained from 204 patients for whom prostate biopsy results are available. Each patient's data includes a variety of MRI sequences, such as T2-weighted images, Diffusion-Weighted Images (DWIs), and Dynamic Contrast Enhanced (DCE) images, alongside annotated clinical findings and Gleason scores.

We randomly divided the dataset into training, validation, and test cohorts with sizes of 154, 50, and 50 respectively, and trained a shallow 3D CNN using DWIs for patient-level prediction of significant Gleason scores, according to methods detailed elsewhere [24], [25]. Unlike previous studies on PROSTATEx, we did not use biopsy coordinates to guide the CNN, which led to comparatively poorer results. However, for the objectives of this study, the level of performance is not a primary concern. The test set predictions are included when the IMICS ROC Analyzer is installed. Fig. 6 displays the ROC plot, SScores, and AUROC contribution for one of the patients analyzed using the IMICS ROC Analyzer, both with and without the exclusion of the top 3 outliers. As demonstrated in Fig. 6, the exclusion of three specific test examples accounts for an 11% decrease in AUROC.

## C. BraTS Dataset

We developed a radiomics-based pipeline to classify low-grade glioma (LGG) and high-grade glioma (HGG) brain tumors using T1-contrast-enhanced (T1CE) MR images. The dataset was sourced from open-radiomics [26] and included 369 adult patients with brain tumors from BraTS 2020 [27]–[29]. The data was split into a stratified 70% training set and 30% test set. On the training set, features with variance below 0.05 were removed, and the Minimum Redundancy Maximum Relevance (mRMR) method was used to select the top 200 out of 1688 radiomics features for classification. The same 200 features were selected from the test set, and an XGBoost classifier [30] was trained. The model achieved an AUROC of 0.949 on the test set. The test set results were used to evaluate the stability of the IMICS ROC Analyzer on larger datasets, as illustrated in Fig. 7. The tool identified six patients as outliers. As an alternative method for outlier identification, we calculated the absolute difference between ground truth labels and predicted probabilities. This difference-based method assigns a score to each patient but does not determine specific outliers without thresholding and fails to rank outliers precisely, as shown in Table II. For example, excluding the last four patients in Table II from the test set does not have an effect on AUROC.

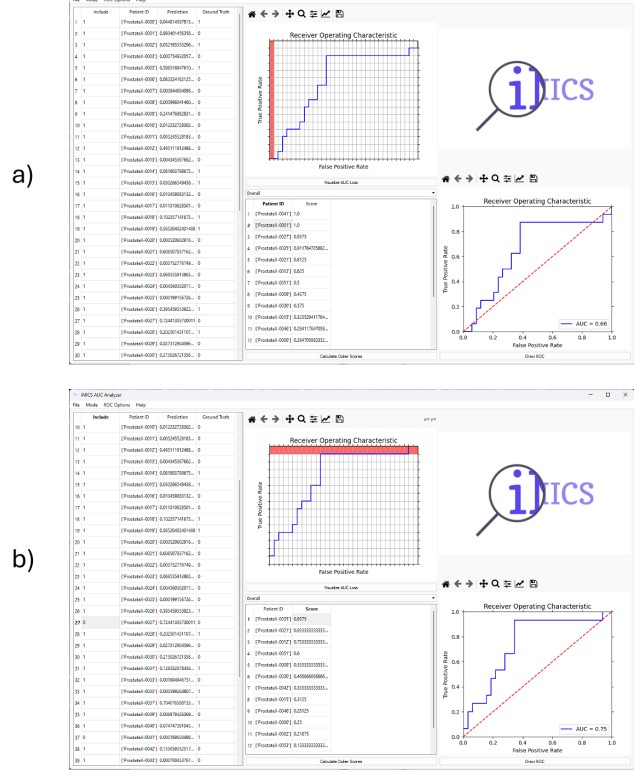

a)

b)

Fig. 6. ROC analysis of ProstateX results: a) without excluding outliers b) with excluding outliers

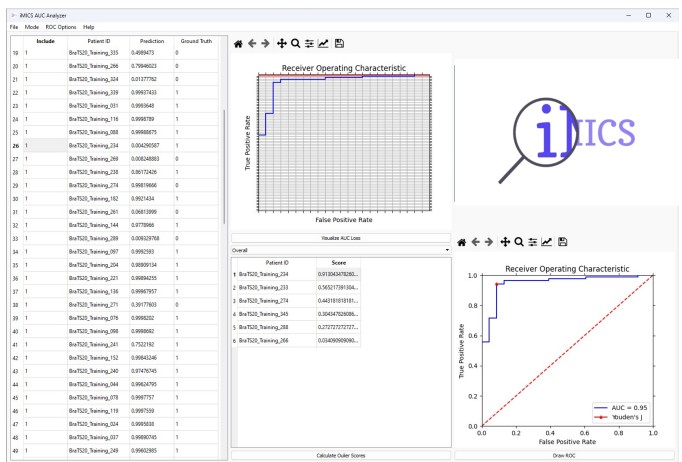

Fig. 7. ROC analysis of BraTS results

TABLE II
DIFFERENCE-BASED VERSUS IMICS ROC ANALYZER OUTLIER SCORES CALCULATION

| Patient ID | Prediction | Ground Truth | Outlier Score | Difference |
|---|---|---|---|---|
| BraTS20_Training_274 | 0.998 | 0 | 0.443 | 0.998 |
| BraTS20_Training_234 | 0.004 | 1 | 0.913 | 0.996 |
| BraTS20_Training_288 | 0.993 | 0 | 0.273 | 0.995 |
| BraTS20_Training_233 | 0.035 | 1 | 0.565 | 0.965 |
| BraTS20_Training_345 | 0.154 | 1 | 0.304 | 0.846 |
| BraTS20_Training_266 | 0.799 | 0 | 0.034 | 0.799 |
| BraTS20_Training_264 | 0.658 | 0 | - | 0.658 |
| BraTS20_Training_335 | 0.499 | 0 | - | 0.499 |
| BraTS20_Training_285 | 0.407 | 0 | - | 0.407 |
| BraTS20_Training_271 | 0.392 | 0 | - | 0.392 |

*D. Non-medical Use Case*

ROC analysis can be applied to any binary classification scenario and is not limited to medicine. Thus, we used a large and imbalanced non-medical dataset to further validate IMICS ROC Analyzer. Specifically, we developed an ML pipeline to classify phishing and non-phishing websites using the Phishing Websites Dataset from the UC Irvine Machine Learning Repository [31]. This dataset comprised 11,055 websites, with 6,157 labeled as phishing and 4,898 as legitimate. For model evaluation, we constructed an imbalanced dataset by randomly sampling 96 phishing examples and 4,800 non-phishing examples. The data was then divided into a stratified

50% training set and 50% test set.

To mitigate class imbalance in the training data, we applied the Synthetic Minority Over-sampling Technique (SMOTE) [32], generating synthetic examples to balance the dataset. Following this preprocessing step, a LightGBM classifier [33] was trained, with hyperparameters optimized via 5-fold cross-validation grid search on the training set, according to Table III. The model achieved an impressive AUROC of 0.965 on the test set, with a 95% confidence interval of [0.918, 0.993]. To facilitate the reproducibility of our results, the dataset and test outcomes have been made publicly available on our GitHub repository.

In addition, we employed the IMICS ROC Analyzer to evaluate the tool's capability in handling datasets containing thousands of examples, whether medical and non-medical. The tool successfully computed outlier scores, and notably, the exclusion of the highest-ranked outlier (ID=2864) resulted in a 2% improvement in AUROC. However, we observed challenges in visualizing AUROC loss on ROC plots for such large datasets, primarily due to the granularity of the grid and the minimal size of the lost AUROC area (Fig. 8). Nevertheless, IMICS ROC Analyzer did not fail to visualize the AUROC loss.

TABLE III
HYPERPARAMETER GRID SEARCH FOR LIGHTGBM CLASSIFIER

| Hyperparameter | Values |
|---|---|
| num_leaves | {31, 50, 100} |
| max_depth | {-1, 10, 20} |
| learning_rate | {0.01, 0.1, 0.2} |
| min_data_in_leaf | {20, 50, 100} |
| scale_pos_weight | {1, all/phishing ratio, 10} |

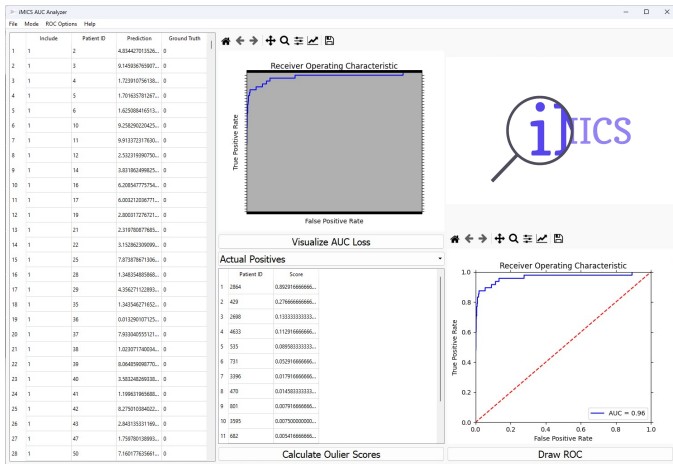

Fig. 8. ROC analysis of the non-medical use case results

## V. DISCUSSION

Equations 1 and 2 imply that TPR and FPR are only affected by actual positives and actual negatives, respectively. As a result, APs manifest on the vertical axis of the ROC curve, whereas ANs delineate the horizontal axis.

Upon derivation of the ROC curve, computation of the AUROC is achieved through integration of the curve between the limits of 0 and 1. It has been elucidated that the AUROC can be conceptualized independent of the ROC curve [19]. Under the premise that a randomly selected positive case and a randomly selected negative case are evaluated, the AUROC is the probability that the positive case will receive a higher prediction score than the negative case. This indicates that AUROC is a measure of goodness of ranking. This conceptual framework enables the derivation of the ROC curve directly from the AUROC calculation algorithm. Namdar et al. provide a pseudocode to define AUROC as a probability and use nested for loops to compare negative and positive examples [19]. Inspired by this algorithm, we proposed a methodology to quantify the AUROC loss contribution of individual examples within an evaluated cohort.

In this paper, we propose shifting from traditional population-based ROC analysis, which typically reports overall AUROC, to a patient-centric ROC analysis approach. This method addresses two key questions for individual patients: (A) Which patients are outliers for the classifier? (B) How much does excluding an outlier affect the ROC/AUROC? This patient-centric perspective aligns the analysis pipeline more closely with the principles of precision medicine, allowing for a sophisticated evaluation of classifier performance and its impact on individual patient predictions.

An ROC curve is derived through plotting TPR and FPR across the thresholds and is on a grid of AP equal-height rows on the vertical axis and AN equal-width columns on the horizontal axis. Such grid, which is used by our proposed

Python functions, compared with an arbitrary one which is usually used in the existing libraries, will streamline extracting the coordinates of any point on the curve (e.g., B point).

In this study, we propose the B point as a replacement for the J point [5] in determining the optimal threshold on a ROC curve. The J point is identified where the summation of sensitivity and specificity is maximized. However, the J point may not be unique and does not consistently represent a balanced point on the ROC curve. In contrast, the B point is always unique and represents the optimal balance point on any ROC curve, especially useful when aiming to balance the weights of positive and negative classes in an imbalanced dataset.

The core principles underlying AUROC loss contribution, outlier scoring, and B point are universally applicable, making the plotting function suitable for a wide range of scenarios. Despite this, the IMICS ROC analyzer, currently in its beta iteration, does not incorporate this plotting function. Furthermore, it necessitates the manual exportation of model predictions and ground truth labels into Comma-Separated Values (CSV) format, and it does not offer an Application Programming Interface (API) for integration with ML environments. Lastly, IMICS ROC analyzer as an installable software is only available for Windows platform, and Linux or Mac users need to run the Python source. As an alternative approach, Linux or Mac users can use the Wine package (https://www.winehq.org/) to run IMICS ROC analyzer. IMICS ROC analyzer can help identify potential bias in the classifiers, but it will not mitigate the bias. Any bias in input may manifest in the output of IMICS ROC analyzer. As future work, we will apply our proposed method to improve a downstream task such as outlier identification in the training set for active learning.

## ACKNOWLEDGMENT

This work was supported by VinBrain JSC and the Chair in Medical Imaging and Artificial Intelligence, a joint Hospital-University Chair between the University of Toronto, The Hospital for Sick Children, and the SickKids Foundation.

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
