# OpenReview forum: "Advanced Receiver Operating Characteristic Curve Analysis to Identify Outliers in Binary Machine Learning Classifications for Precision Medicine"
_IEEE.org/EMBS/BHI/2024/Conference — IEEE BHI'24_

### Official Review · Reviewer_igts · 2024-08-10
**Advanced Receiver Operating Characteristic Curve Analysis to Identify Outliers in Binary Machine Learning Classifications for Precision Medicine**

**Overall Rating:** 7
**Confidence:** 4

**Other Quality Metrics:**

(a) Clarity of writing; Fair
(b) Clinical Significance; Fair
(c) Methodological Novelty; Fair
(d) Experiments and Results; Fair

**Questions For The Authors:**

Expand Dataset Testing: Test the proposed methods on a wider variety of datasets, including those from non-medical domains, to validate their generalizability and robustness across different applications.
Investigate Impact on Class Imbalance: Conduct further studies to explore how the proposed methods handle class imbalance in binary classification tasks, potentially adapting the scoring system to account for imbalanced data distributions.

**Strengths:**

Innovative Approach: The introduction of a novel scoring method to quantify the contribution of individual data points to AUROC loss is a significant advancement in ROC curve analysis, providing deeper insights into classifier performance.
Practical Application: The IMICS ROC Analyzer offers a practical tool for researchers and clinicians to perform detailed ROC analysis, which is particularly valuable in precision medicine.
Comprehensive Evaluation: The application of the proposed methods to both synthetic and real-world datasets demonstrates their robustness and applicability across different domains, particularly in medical diagnostics.
Focus on Precision Medicine: By allowing for patient-centric ROC analysis, the paper aligns with the growing emphasis on precision medicine, enabling more personalized and accurate assessments of machine learning classifiers.

**Summary Of The Paper:**

This paper introduces an advanced method for Receiver Operating Characteristic (ROC) curve analysis, specifically designed to identify outliers in binary machine learning classifications. The study focuses on improving the accuracy and reliability of ROC analysis by introducing a novel scoring method to rank actual positives and negatives based on their contribution to the Area Under the ROC Curve (AUROC) degradation. The paper also presents the IMICS ROC Analyzer, a graphical user interface (GUI)-based software tool that implements these advanced ROC analysis techniques. The proposed methods are evaluated using a synthetic ROC curve and real-world datasets, including a prostate cancer dataset from the PROSTATEx challenge and the BraTS dataset for brain tumor classification. The results demonstrate the effectiveness of the proposed methods in identifying outliers and refining the ROC curve analysis to enhance the precision of machine learning classifiers in medical applications.

**Weaknesses:**

Generalizability of Results: The method has been tested on a limited number of datasets, primarily from the medical domain. Testing on a broader range of datasets, including those from non-medical fields, would strengthen the generalizability of the proposed approach.
Impact on Class Imbalance: The study focuses on binary classification tasks but does not address how the proposed methods perform in scenarios with significant class imbalance. Further analysis in such contexts would provide a more complete understanding of the method's effectiveness.

---

### Official Review · Reviewer_R8st · 2024-08-12
**Advanced Receiver Operating Characteristic Curve Analysis to Identify Outliers in Binary Machine Learning Classifications for Precision Medicine**

**Overall Rating:** 7
**Confidence:** 3

**Other Quality Metrics:**

(a) Clarity of writing: good
(b) Clinical Significance: fair
(c) Methodological Novelty: good
(d) Experiments and Results:good

**Questions For The Authors:**

no

**Strengths:**

The proposed IMICS ROC Analyzer enhances the field of precision medicine by allowing for measuring an individual’s contributions to the overall AUROC, thus facilitating confidence measurement of Machine Learning (ML) classifiers for individual cases of interest in a cohort.

**Summary Of The Paper:**

The authors proposed a novel scoring method to rank actual positives and actual negatives within a test set, according to their impact on AUROC degradation. Furthermore, the authors introduced the IMICS ROC Analyzer, a graphical user interface (GUI)-based software embedded with their innovative algorithms.

**Weaknesses:**

no

---

### Decision · Program_Chairs · 2024-09-23

Accept